# Cognitive Impairment Mediates the Association between Dietary Inflammation and Depressive Symptoms in the Elderly

**DOI:** 10.3390/nu14235118

**Published:** 2022-12-02

**Authors:** Mengzi Sun, Ling Wang, Yueyang Hu, Xuhan Wang, Shoumeng Yan, Yinpei Guo, Jing Li, Zechun Xie, Bo Li

**Affiliations:** 1Department of Epidemiology and Biostatistics, School of Public Health, Jilin University, Changchun 130021, China; 2Department of Social Medicine and Health Management, School of Public Health, Jilin University, Changchun 130012, China; 3School of Nursing, Jilin University, Changchun 130021, China

**Keywords:** cognitive function, dietary inflammatory index, depressive symptoms, mediation

## Abstract

The underlying mechanism in both cognitive impairment and depression was chronic inflammation, which could be reflected by the dietary inflammatory index (DII). However, the effect of cognitive impairment on the association between DII and depression was not clear. Therefore, in this study, we hypothesized that cognitive impairment could mediate the association between dietary inflammation and depressive symptoms. A total of 2550 participants aged ≥60 from the National Health and Nutrition Examination Survey (NHANES) in 2011–2014 were involved in the serial, cross-sectional study. Proinflammatory and anti-inflammatory diets were measured by DII. Cognitive impairment was measured by four dimensions, CERAD-immediate, CERAN-delayed, animal fluency test, and DSST. Depressive symptoms were measured by PHQ-9 scores. We found that a proinflammatory diet and cognitive impairment were both risk factors for depressive symptoms. An interaction between an inflammatory diet and cognitive impairment was detected (*P*-interaction = 0.060). In addition, all four dimensions of cognition mediated the association between DII and depressive symptom scores. Part of the association between DII and depressive symptoms scores could be explained by different dimensions of cognitive function, and the proportion of mediation ranged from 10.0% to 36.7%. In conclusion, cognitive impairment levels partly mediated the association between DII and depressive symptoms.

## 1. Introduction

Depression is a common mental disorder characterized by a depressed mood, lack of energy, sadness, insomnia, and an inability to enjoy life [1]. It was reported that the estimated prevalence of depression was 20.04%, including 14.60% for mild depression, 3.70% for moderate depression, and 1.74% for severe depression among adults aged 65 years and over in the U.S., and the trend of severe depression in the elderly was increasing [2]. The previous study has suggested that the risk factors for depression in the elderly include being female, social isolation, uncontrolled pain, insomnia, and functional and cognitive impairment [3].

Cognitive function is a mental process of knowledge acquisition, including the response to learning, concentration, memory, and decision-making [4]. A U.S. study reported that approximately one-third of individuals aged 65 years and older had dementia or mild cognitive impairment (MCI) [5], which is a psychiatric syndrome characterized by a cognitive decline greater than that expected based on an individual’s age and education level [6]. Moreover, subjects with MCI were more likely to develop depression compared with those with normal cognitive function, especially with those with amnestic MCI [7]. Previous studies also reported that populations with both MCI and depression typically had lower processing speeds and showed a decrease in executive function, flexibility, and lexico-semantic function than MCI patients without depression [8]. Furthermore, a review indicated that cognitive dysfunction mediated psychosocial and day-to-day impairment in depressed individuals [9].

Previous studies have proved the positive association between a proinflammatory diet and depressive symptoms [10,11]. It was also reported that individuals with depressive disorders exhibit increased levels of inflammatory markers, such as C-reactive protein [12]. Chronic, low-level inflammation seems to trigger depression via a multitude of mechanisms [13]. Furthermore, the low degree of systemic proinflammation in adipose tissue circulation and neuroinflammation in the brain hippocampus may be related to neurocognitive dysfunction [14]. In the brain, in particular, proinflammatory conditions can activate astrocytes and microglia, causing chronic inflammatory damage and leading to progressive neuronal damage [15]. Noteworthily, previous studies indicated that dietary inflammatory potential might be the bridge linking diet and chronic disease [16,17]. The dietary inflammatory index (DII) has been developed to evaluate the inflammatory potential, which might be the “bridge” we mentioned above, and is a literature-derived score [18]. Meanwhile, DII could quantify the evaluation of the relationships between diet and health outcomes [19].

Multiple studies indicated a significant association between DII and both depression and cognitive impairment [20,21]. However, the relationship between DII, cognitive impairment, and depression has not been illustrated. In this study, we hypothesized that cognitive impairment mediates the association between DII and depressive symptoms in the elderly, and we explored these associations based on the National Health and Nutrition Examination Survey (NHANES).

## 2. Materials and Methods

### 2.1. Sample

The NHANES, which were conducted by the Centers for Disease Control and Prevention (CDC), aimed to assess the health status of the U.S. non-institutionalized civilian population. This survey utilized a complex probability sampling design and collected information through standardized interviews, physical examinations, and tests of biological samples [22]. A total of 19,931 subjects were enrolled in NHANES in 2011~2014. A total of 3632 subjects were aged 60 years and over. Moreover, 698 participants did not complete the cognitive functional questionnaire and 50 participants did not complete the depression screening instrument. Meanwhile, 334 participants were excluded due to missing or extreme diet data (total energy intakes of <500 or >5000 kcal/day for females and <500 or >8000 kcal/day for males) and missing data on BMI, WC, smoking, drinking, and physical activity data. Finally, a total of 2550 participants were involved in the study (Figure 1). Additionally, based on the G power, we set the effect size f = 0.4, total sample size = 2550, number of groups = 2, and number of covariates = 12, then we get a power equal to 1.0, which indicated that our sample size was acceptable for the result.

### 2.2. Data Measurement

#### 2.2.1. Outcome Ascertainment

Depressive symptoms were assessed with a short screening questionnaire and a patient health questionnaire (PHQ-9), which is a valid scale for assessing depressive symptoms based on the Diagnostic Statistical Manual of Mental Disorders (DSM)-V. It scores the 9 items from “0” (not at all) to “3” (nearly every day) [23]. All these question scores are then summed to calculate the overall depressive symptom scores for every subject in the study (range 0–27). Previous studies have shown the validity of PHQ-9 scores ≥ 10 to define depressive symptoms (sensitivity: 88%, specificity: 88%) [23]. Therefore, participants were divided into no depressive symptoms (PHQ-9 score < 10) and depressive symptoms (PHQ-9 score ≥ 10).

#### 2.2.2. Exposure Measurement

##### Dietary Inflammatory Index (DII)

The dietary data were obtained by the average of two interviews of the above, and if participants were interviewed only on the first day, the dietary data on the first day was selected.

The DII, developed by Shivappa through a literature review, was used to evaluate the potential inflammatory levels of dietary components [18]. DII is based on hundreds of peer-reviewed articles linking any aspect of diet to at least one of six inflammatory biomarkers: IL-1β, IL-4, IL-6, IL-10, TNF-α, and C-reactive protein (CRP). The detailed algorithm has been described in other studies. In this study, 27 nutrients were used for the calculation of the DII, which include alcohol, vitamin B12/B6, β-carotene, caffeine, carbohydrate, cholesterol, total fat, fiber, folic acid, Fe, Mg, Zn, Se, MUFA, niacin, n-3 fatty acids, n-6 fatty acids, protein, PUFA, riboflavin, saturated fat, thiamin, and vitamins A/C/D/E. Importantly, even if the number of nutrients applied for the calculation of DII is <30, the DII is still available [18]. It was reported that positive scores represented a proinflammatory diet and negative scores represented an anti-inflammatory diet. We then divided the participants into an anti-inflammatory diet (DII < 0) group and a proinflammatory diet (DII ≥ 0) group [24].

##### Cognitive Function

Only those over 60 years of age participated in the survey. There were four dimensions of cognitive function definition. The Consortium to Establish a Registry for Alzheimer’s Disease (CERAD) word learning test assessed the learning ability for new verbal information, including immediate and delayed. The animal fluency test was used to examine categorical verbal fluency in executive function. The digit symbol substitution test (DSST) was used to assess processing speed, sustained attention, and working memory. Cognitive impairment scores were the sum of all four dimensions, whose lowest 25th percentile was assigned as 1 point and other quartiles were assigned as 0 points [25]. The higher the score, the more severe the level of cognitive impairment.

#### 2.2.3. Covariate Assessment

Smoking status was defined in three categories: Non-smokers (who never had at least 100 cigarettes in their lifetime); former smokers (who had at least 100 cigarettes but did not smoke now); Current smokers (who had at least 100 cigarettes and reported the number of cigarettes per day in the past 30 days) [26].

Drinking status was defined in three categories: Non-drinkers (who did not have at least 12 alcohol-based drinks in the past year or lifetime); former drinkers (who had at least 12 drinks in their lifetime but not in the past year); current drinkers (who had at least 12 drinks in the past year and reported the number of drinks per week) [26].

Physical activity was defined as two groups according to the Global Physical Activity Questionnaire and analyzed according to the WHO guidelines [27]. The active group was defined as those who participated in more than 149 min of moderate physical activity, more than 74 min of vigorous physical activity, or more than 599 metabolic equivalents (MET)-minutes per week. Another population was defined as an inactive group. Moreover, arthritis, congestive heart failure, and stroke were identified by the question “Ever been told you have this disease by doctors?”

### 2.3. Statistical Analysis

Continuous variables and categorical variables were described as mean and standard error (SE), unweighted frequency, and weighted percentage, respectively. *t*-test and chi-square test were performed for the comparison of continuous and categorical variables, respectively. Binary logistic regression was used to analyze the association among depressive symptoms, cognitive impairment, and inflammatory diet under the adjustment of confounders. The sensitivity analysis of this study was achieved through stratified analysis, and P-interaction between cognitive impairment levels and each stratified variable was also tested. Then, to investigate whether the cognitive impairment levels mediated the association between DII and depressive symptoms, three pathways (a, b, and c) were used to assess the mediation [28] (Figure 2). The total effect evaluated the association between DII (exposure) and depressive symptoms (outcome). Path a assessed the association between DII and each dimension of cognitive function (mediator). Path b measured the association between each dimension of cognitive function (mediator) and depressive symptoms (outcome). Path c was the total (and direct) effect of DII and depressive symptoms, and path c’ was the direct effect of DII and depressive symptoms. The influence of the four dimensions of cognitive function on the link between DII and depressive symptoms was assessed through path c’ (direct effect). The proportion of the mediated effect was calculated using the following formula: (mediated effect/total effect) × 100%. Bootstrapping was used for significance testing for the mediation analysis.

All statistical analyses were conducted using IBM SPSS 26.0, Mplus 8.3, and R version 4.1.0, and the package “survey” [29] was used. A 2-sided *p*-value less than 0.05 was considered significant, and *P*-interaction less than 0.1 was considered significant.

### 2.4. Ethics Approval and Consent to Participate

The protocols of NHANES were approved by the institutional review board of the National Center for Health Statistics, CDC. Written informed consent was obtained from each participant before participation in this study.

## 3. Results

This study involved a total of 2550 participants, including 1226 participants with an anti-inflammatory diet and 1324 participants with a proinflammatory diet (Table 1). Compared with the anti-inflammatory diet group, the levels of BMI and cognitive impairment level of subjects with a proinflammatory diet were higher (*p* < 0.05), and the scores of CERAD-immediate, CERAD-delayed, Animal Fluency test, DSST, and energy intake were lower (*p* < 0.05). The proportion of participants with depressive symptoms was higher in the proinflammatory diet group than that in the anti-inflammatory diet group.

Based on the results of Table 2, we found that an inflammatory diet and cognitive impairment were positively associated with the risk of depressive symptoms (Model 1). After controlling for covariates (Model 2 and Model 3), the association of cognitive impairment with depressive symptoms became even stronger. Especially, in the final model, compared with the anti-inflammatory group, the OR of the proinflammatory diet was 2.86 (1.01, 8.11), and compared with the non-cognitive impairment group, the OR of the cognitive impairment level was 1.36 (1.16, 1.59) of the depressive symptoms, respectively.

Table 3 shows the results of a stratified analysis of the association between cognitive impairment level and depressive symptoms. Except for dietary inflammation (*P*-interaction = 0.060), all subgroups had no interaction of cognitive impairment level on depressive symptoms, and a proinflammatory diet was a risk factor for depressive symptoms in each subgroup. The odds ratio of cognitive impairment level on depressive symptoms was higher in the anti-inflammation group than that in the proinflammation group.

Then, we explored the mediation effect of the four dimensions of cognitive impairment on the association between DII and depressive symptoms scores, as shown in Table 4. CERAD-immediate, CERAD-delayed, animal fluency test and DSST partly mediated this association, and the proportion ranged from 10.0% to 36.7%.

## 4. Discussion

In the present study, the associations among inflammatory diet, cognitive impairment, and depressive symptoms were explored in the elderly based on NHANES. The elderly with a proinflammatory diet or cognitive impairment were found to be more likely to have depressive symptoms. Interestingly, there was an interaction between an inflammatory diet and cognitive impairment level on depressive symptoms. Different dimensions of cognitive impairment mediated the association between DII and depressive symptoms to varying degrees, and the proportion ranged from 10.0% to 36.7%. To our knowledge, this is the first study to focus on the associations between DII, cognitive impairment, and depressive symptoms in the elderly.

A cohort study evaluating the association between DII and cognitive performance 13 years later in middle-aged people showed that a proinflammatory diet was negatively related to cognitive function [30]. It was indicated that an individual with an inflammatory imbalance due to diet or inflammatory conditions may worsen cognitive decline [31]. Another study also found that DII was associated with DSST, including worse episodic memory, working memory, and semantic memory [32]. These findings indicated that dietary inflammation, assessed via the DII, may be less related to memory encoding, but may have a larger influence on memory consolidation, which was consistent with our study [32]. The hippocampus, a subcortical memory structure, receives inputs from brainstem feeding areas [33], and a review suggested the links between the hippocampus and peripheral and central processing of eating-related signals which support its role in the control of food intake [34].

The previous meta-analysis indicated that there were moderate cognitive deficits in the population with depressive symptoms involved in executive function, memory, and attention [35]. The individuals experiencing both MCI and depression have difficulty with immediate and delayed memory tasks in comparison with non-depressed persons with MCI [36]. Besides, it was suggested that late-life major depression is often associated with peripheral body changes and cognitive impairment [37]. An Irish population study found that DII was positively associated with depressive symptoms in females [38], which was consistent with our study. The mechanism may be through the activation of the innate immune system via corresponding proinflammatory nutrients, leading to low-grade inflammation and chronic diseases, including mental health disorders [39]. A meta-analysis showed that the prevalence of depressive symptoms was 32% in patients with MCI [36]. Furthermore, the comorbidity of depressive symptoms and cognitive impairment could be an intervention target for the prevention of dementia in MCI subjects [40].

Our study found that a significant association between a proinflammatory diet and depressive symptoms was not found in the cognitive impairment patients, but the association existed in the normal group. Diet could be promoting inflammation and also could be decreasing inflammation, depending on their hormonal response [41]. The persistent activation of the immune system might be caused by a proinflammatory diet, leading to low-grade inflammation [42]. Dietary inflammation was regarded as a preventable function of poor diet [32].

Moreover, there is indeed evidence that potentially warrants the subcategorization of depressed patients into inflammatory and non-inflammatory subtypes, and anti-inflammatory drugs will display high efficacy in both subtypes [13]. Therefore, decreasing inflammation might be a benefit for depressive symptoms, and our study suggested that reducing dietary inflammation is important in reducing the risk of depressive symptoms by reducing the risk of cognitive impairment to a certain extent, especially at the stage when the diet has a greater impact on the depressive symptoms.

The present study has several strengths. To the best of our knowledge, this is the first study to assess the mediating effect of different dimensions of cognitive impairment on the association between DII and depressive symptoms scores in a nationally representative sample of the elderly in America. Moreover, NHANES used rigorous data collection procedures to collect information from a large number of participants to generate nationally representative estimates of American older adults. However, the present study also has some limitations. Firstly, it was a cross-sectional design, precluding the establishment of causality, and the relevant variables were self-reported, which might introduce recall bias leading to over- or under-estimation of the association between DII and depressive symptoms. Secondly, it was important to consider the difficulty of an accurate dietary evaluation with possible underreporting or misclassification. We excluded all the missing values instead of imputation, which might reduce the representativeness of the samples. Thirdly, there were only 27 nutrients to calculate the DII instead of the original 45, which might have ignored the consumption of the other 18 nutrients.

In conclusion, our results suggested that the association between DII and depressive symptoms was mediated to varying degrees by different dimensions of cognitive impairment. Further research is essential to confirm these results in a prospective study for older American adults to explore the mechanism of the positive associations.

## 5. Conclusions

There was an interaction between an inflammatory diet and cognitive impairment in depressive symptoms. Moreover, the CERAD-immediate test, CERAD-delayed test, animal fluency test, and DSST test partly mediated the association between DII and depressive symptoms.

## Figures and Tables

**Figure 1 nutrients-14-05118-f001:**
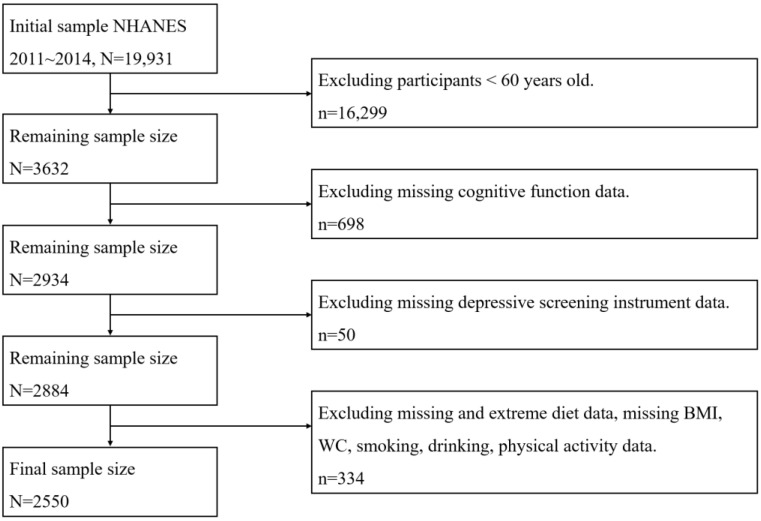
Flowchart for the study design and participants.

**Figure 2 nutrients-14-05118-f002:**
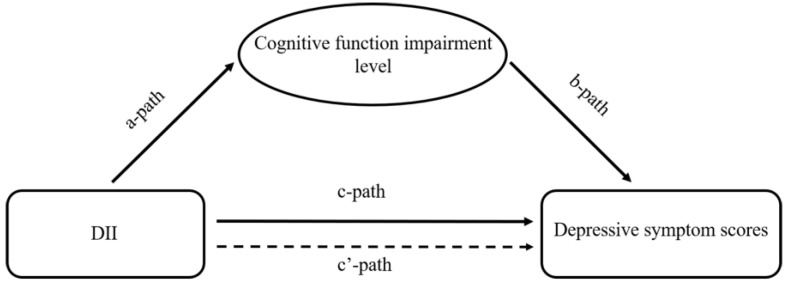
Path diagram of the mediation analysis models.

**Table 1 nutrients-14-05118-t001:** Characteristics of the participants with an anti-inflammatory diet and proinflammatory diet (Mean (SE)/N(%)).

Characteristics		Anti-Inflammatory Diet (N = 1226)	Proinflammatory Diet (N = 1324)	*χ*^2^/t	*p*
Age (Mean (SE)) ^a^		69.15 (0.33)	68.79 (0.31)	−0.923	0.363
BMI (Mean (SE)) ^a^		28.56 (0.34)	29.52 (0.34)	2.210	0.035
WC (Mean (SE)) ^a^		101.86 (0.94)	103.16 (0.66)	1.237	0.226
Cognitive impairment level (Mean (SE)) ^a^		0.59 (0.04)	0.89 (0.05)	5.009	<0.001
CERAD-immediate (Mean (SE)) ^a^		6.74 (0.09)	6.45 (0.09)	−3.187	0.003
CERAD-delayed (Mean (SE)) ^a^		6.49 (0.12)	6.19 (0.14)	−2.088	0.045
Animal Fluency test (Mean (SE)) ^a^		19.22 (0.26)	17.13 (0.24)	−6.596	<0.001
DSST (Mean (SE)) ^a^		55.26 (0.83)	50.17 (0.79)	−4.861	<0.001
Energy intake (Mean (SE)) ^a^		2239.42 (23.10)	1552.70 (26.53)	−21.439	<0.001
Gender (N (%)) ^b^				51.078	<0.001
	Male	688 (53.85)	572 (39.71)		
	Female	538 (46.15)	752 (60.29)		
Race (N (%)) ^b^				25.475	<0.001
	Mexican American	117 (3.36)	112 (4.04)		
	Other Hispanic	116 (3.65)	150 (4.91)		
	Non-Hispanic White	647 (80.45)	598 (75.50)		
	Non-Hispanic Black	220 (6.18)	368 (10.92)		
	Other race	126 (6.36)	96 (4.63)		
Smoking status (N (%)) ^b^				34.860	<0.001
	Never	599 (49.86)	646 (51.98)		
	Current Smoker	109 (6.95)	209 (12.99)		
	Formal Smoker	518 (43.19)	469 (35.04)		
Drinking status (N (%)) ^b^				68.042	<0.001
	Never	150 (8.62)	225 (17.40)		
	Current drinker	917 (80.08)	858 (66.05)		
	Formal drinker	159 (11.30)	241 (16.55)		
Physical activity (N (%)) ^b^				13.257	0.021
	Inactive	550 (43.13)	716 (50.33)		
	Active	676 (56.87)	608 (49.67)		
Depressive symptoms (N (%)) ^b^				29.516	0.001
	No	1138 (95.35)	1185 (89.74)		
	Yes	88 (4.65)	139 (10.26)		
Stroke (N (%)) ^b^				1.128	0.388
	No	1153 (94.76)	1239 (93.79)		
	Yes	73 (5.24)	88 (6.21)		
Arthritis (N (%)) ^b^				0.370	0.772
	No	648 (49.13)	672 (50.34)		
	Yes	578 (50.87)	652 (49.66)		
Congestive heart failure (N (%)) ^b^				1.171	0.407
	No	1162 (94.15)	1223 (93.10)		
	Yes	64 (5.85)	101 (6.90)		

^a^ for continuous variables compared by *t*-test; ^b^ for categorical variables compared by chi-square test.

**Table 2 nutrients-14-05118-t002:** Logistic regression model for the association of Inflammatory diet and Cognitive impairment level on depressive symptoms.

	Model 1		Model 2		Model 3	
	OR (95% CI)	*p*	OR (95% CI)	*p*	OR (95% CI)	*p*
Inflammatory diet (reference = Anti-inflammatory)	2.19 (1.23, 3.89)	0.009	2.82 (0.98, 8.14)	0.055	2.86 (1.01, 8.11)	0.048
Cognitive impairment level (reference = No)	1.23 (1.05, 1.46)	0.013	1.40 (1.19, 1.64)	<0.001	1.36 (1.16, 1.59)	<0.001

Model 1 = Inflammatory diet + Cognitive impairment level. Model 2 = Model 1 + Gender + Age + Race + BMI + WC + Smoking status + Drinking status + Physical activity + Energy intake. Model 3 = Model 2 + Stroke + Arthritis + Congestive heart failure.

**Table 3 nutrients-14-05118-t003:** Stratified analyses of the associations between cognitive impairment level and depressive symptoms.

Variables		OR	95% CI	*p*	*P*-Interaction
Gender					0.119
	Male (N = 1262)	1.68	1.29–2.18	<0.001	
	Female (N = 1291)	1.19	0.98–1.44	0.076	
Race					0.110
	Mexican American (N = 229)	1.12	0.84–1.50	0.438	
	Other Hispanic (N = 266)	1.66	1.20–2.30	0.003	
	Non-Hispanic White (N = 1246)	1.25	0.94–1.67	0.117	
	Non-Hispanic Black (N = 590)	1.88	1.30–2.72	0.001	
	Other race (N = 222)	2.56	1.51–4.35	0.001	
Physical exercise					0.744
	Inactive (N = 1269)	1.45	1.21–1.73	<0.001	
	Active (N = 1284)	1.26	0.87–1.81	0.219	
Smoking status					0.169
	Non-smoker (N = 1247)	1.25	0.97–1.60	0.078	
	Current smoker (N = 318)	0.92	0.64–1.33	0.656	
	Former smoker (N = 988)	1.75	1.38–2.21	<0.001	
Drinking status					0.113
	Non-drinker (N = 375)	1.49	1.04–2.13	0.030	
	Current drinker (N = 1775)	1.39	1.20–1.62	<0.001	
	Former drinker (N = 400)	1.75	1.10–2.79	0.019	
Inflammatory Diet					0.060
	Proinflammation (N = 1327)	1.32	1.04–1.67	0.023	
	Anti-inflammation (N = 1226)	1.50	1.21–1.85	<0.001	
Stroke					0.479
	Yes (N = 161)	1.59	1.02–2.46	0.039	
	No (N = 2392)	1.36	1.12–1.65	0.003	
Arthritis					0.639
	Yes (N = 1230)	1.29	1.05–1.58	0.016	
	No (N = 1323)	1.38	1.08–1.75	0.010	
Congestive heart failure					0.116
	Yes (N = 165)	0.69	0.37–1.29	0.235	
	No (N = 2388)	1.45	1.23–1.71	<0.001	

Adjusted for gender, age, race, BMI, WC, smoking status, drinking status, physical activity, energy intake, stroke, arthritis, and congestive heart failure. Of note, the variables examined in this table were not adjusted.

**Table 4 nutrients-14-05118-t004:** Mediation effect of the four dimensions on the association between DII and depressive symptoms scores (β(SE) and *P*/95% CI).

Mediator	Exposure to Mediator	Mediator to Outcome	Direct Effect	Mediated (Indirect Effect)	Total Effect (Exposure to Outcome)	Proportion Mediated (%)
CERAD-immediate	−0.079 (0.019) *p* < 0.001	−0.329 (0.059) *p* < 0.001	0.233 (0.057) *p* < 0.001	0.026 (0.008) 95% CI (0.011, 0.044) *p* = 0.002	0.259 (0.057) *p* < 0.001	10.0
CERAD-delayed	−0.158 (0.029) *p* < 0.001	−0.170 (0.040) *p* < 0.001	0.232 (0.058) *p* < 0.001	0.027 (0.009) 95% CI (0.012, 0.046) *p* = 0.002	0.259 (0.057) *p* < 0.001	10.4
Animal fluency test	−0.490 (0.070) *p* < 0.001	−0.089 (0.013) *p* < 0.001	0.215 (0.058) *p* < 0.001	0.044 (0.010) 95% CI (0.026, 0.066) *p* < 0.001	0.259 (0.057) *p* < 0.001	17.0
DSST	−2.014 (0.205) *p* < 0.001	−0.047 (0.006) *p* < 0.001	0.163 (0.058) *p* = 0.005	0.095 (0.015) 95% CI (0.068, 0.126) *p* < 0.001	0.259 (0.057) *p* < 0.001	36.7

Exposure: DII; outcome: depressive symptoms; adjusted for gender, age, race, BMI, WC, smoking status, drinking status, physical activity, energy intake, stroke, arthritis, and congestive heart failure. Furthermore, 95% CI does not contain 0 representing *p* < 0.05.

## Data Availability

Data described in the manuscript, code book, and analytic code will be made publicly and freely available without restriction at [https://www.cdc.gov/nchs/nhanes (accessed on 22 June 2022)].

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
