# Peer review of "Cognitive Impairment Mediates the Association between Dietary Inflammation and Depressive Symptoms in the Elderly"

_nutrients, 2022, doi:10.3390/nu14235118_

Round 1
Reviewer 1 Report
The manuscript is trying to explore how cognitive impairment mediates the association of dietary inflammation with depressive. The quality and scientific soundness of the presented results is limited. Methodology of performed mediation analysis is unclear and not described in detail. English is poor. Figures and tables are not comprehensive and contain mistakes. Although, the introduction explains in a clear and coherent manner the background of the study, the results did not fully explain the role of mediation of cognitive impairment in the association of dietary inflammation with depressive. The results have rather preliminary character and are not convincing. Below I present only a few my remarks:
1. “Importantly, even if the nutrients applied for the calculation of DII is <30, the DII is still available [18].” Technically DII is “available” but could have limited value. Why did the study include only 30 foods and nutrients to calculate DII instead of original 45? Was this related to availability of variables in the food composition database? What about consumption of eugenol, ginger, saffron, Se, trans fat, turmeric, pepper, thyme/oregano, rosemary and polyphenols (flavan-3-ol, flavones, flavonols, flavonones, anthocyanidins, isoflavones)? How no information about these nutrients could influence DII?
2. Details of methodology for mediation analysis is needed Please add the literature.
3. PHQ-9 is a valid criteria instrument based on DSM-V. – what is “DSM-V”?
4. Please give a reference to cut-off point 0 of DII (anti-inflammatory diet vs pro-inflammatory). Was it suggested in Shivappa? People divide DII e.g. according the tertiles – highest and lowest could define pro-/anti- inflammatory diet
5. Previous studies indicated that 10 points is the cut off value of depressive symptoms (sensitivity:88%, specificity:88%) – impresize – the “optimal” cut-off point?
6. Page 4 “Sensitive analysis of logistic regression was conducted via stratified analysis” – please reformuate this sentence.
7. Difference between c and c’ path in the diagram is unclear and not explained
8. Table 2 Inflammatory diet reference category “No” was not introduced previously – only “pre” “anty”
9. “Specially, in the final model, compared with normal group, the odds of pro-inflammatory diet and cognitive impairment level were 2.86 (1.01, 8.11) and 1.36 (1.16, 1.59) on the depressive symptoms, respectively” Please reformulate this sentence – it is difficult to understand the meaning in this form What’s more, what do you mean by “normal group”
10. Why Table 2 shows Logistic regression model for interaction between Inflammatory diet and Cognitive impairment level on depressive symptoms? I could not find any interaction model in this table. Only independent effect both variables was tested.
11. “Table 3 was the stratified analysis” – Table 3 is just a table not analysis
12. Table 3 looks strange – in separate column just bracket – please use the same style in all tables
13. In Column “Mediated (undirect effect)” of Table 4 what does it mean “95% CI”? Where is p-value? Please correct table.
14. Please improve style of the text of the article, e.g. – just few examples
- “A total of 2550 participants who aged 60 ..” – mean age in Table 1 about 69.Do you mean all participants have 60 years?
- The detailed situation was shown in Figure 1.
- … is an effective tool for evaluating the situation of depressive symptoms
- Page 1, line 92 symptoms of depressive symptom
- Page 3, line 116 sub-test was tested for learning ability
- “Continues variables” – continuous?
- Page 3, line 102 “All participants in this study were completed 24-hour dietary recall interviews”
- “even if the nutrients applied for the calculation of DII is <30” do you mean “number” of nutrients?
Reviewer 2 Report
This manuscript uses statistical analysis to explore the association between dietary intake, cognition and depression. Depression is assessed using the PHQ-9, which probes factors such as appetite, concentration and slowness of movement and speech. Cognition is assessed by a range of tests including the digit symbol substitution test which requires "processing speed, sustained attention and working memory". One would therefore not be surprised to see that those who achieve scores indicative of depression also achieve scores indicative of poor cognition, or vice versa.
Dietary Inflammatory Index scores were based on two, 24-hour, recall interviews. Those participants deemed to have the anti-inflammatory diet had marginally lower BMI, but considerably higher daily energy intake; this better 'weight control' may be associated with the marginally greater physical activity.
The main tenet of the manuscript is that pro-inflammatory diet is associated with poorer cognition and greater risk of depression. The results support this conclusion . The discussion, however, posits that it is the pro-inflammatory diet that has promoted the cognitive decline and depression, such a conclusion may be too simplistic. For example, depression is associated with poor appetite and weight loss, could the depression be responsible for the poor diet?
I believe that there are other possible explanations for the associations seen, and that other evidence should be sought to support their conclusions. For example, is there any evidence that anti-inflammatory drugs (eg daily low-dose aspirin for anti-platelet aggregation) have any beneficial effects on mood / cognition?
Reviewer 3 Report
Dear Authors,
Thank you for the opportunity to review your manuscript “Cognitive Impairment Mediates the Association of Dietary Inflammation with Depressive Symptoms” The study is investigating the hypothesis; does cognitive impairment mediate the association between dietary inflammation and depressive symptoms?
Initially s would be of value to include in the title in an ageing human population as it is not clear from the introduction to what population you are referring.
Additionally, it would be advisable that a complete revision of both spelling and grammar be conducted by a native English speaker as many of the sentence’s structure are incorrect.
It is difficult to understand if you are referring to Cognitive impairment as the mediator or being the causative relation to dietary inflammation with depression or is it, Dietary inflammation having a causal relationship to depressive symptoms or both. It is not clear. There is no clear statement of the objective of this research.
Introduction:
In the initial sentence, it would be advisable to include that depression and how it relates to one’s daily life as the sentence is very open ended.
Line 33: you state” Age is an independent and important risk factor…………” why is this the case, at what age are you referring, is it by sex or are there any specific stages in one’s life?
Line 37: you state”…………….. equal to 65 years in the U.S.” please confirm why the focus is on the US?? Was this a convenient reference?
Line 38 you state” being female, social isolation, uncontrolled pain, insomnia, and functional and cognitive impairment” using reference 4 that only relates to Iran? What about the rest of the world and why was this reference used?
Line 42 you state” prevalence rate of mild cognitive impairment (MCI) ranges from 10% to 20% in the population aged 65 years and older” please explain what population, what country and what are you referring to as MCI as you have yet to explain what this is?? What cognitive domains are you referring to in relation to MCI??
Line 43 you state” Previous study has indicated that cognitive impairment was existed during depressive episodes and remission” reference 7 is a Meta Analysis and not just a study, this sentence needs to be revised.
Line 46 you state” …..cognitive impairment remains unclear, given that cognitive impairment is a risk factor for poor treatment outcomes in depression, a certain link between cognitive impairment and depression could be existed” Please explain what this is relating to as correlation does not relate to causation.
Line 52 you state” “Previous studies have proved the relationship among poor diet, inflammation, and depressive symptomology” please explain what relationship you are referring to and the elements of what you consider a poor diet to be with respect to the statement made and what symptomatology you are referring to?
Line 59-61 does not seem to fit as a statement as it is no adding to the story you are presenting.
Overall, the introduction needs to b re-written and focus on the DII and depression, MCI and the other issues you wish to explore.
Materials and methods: Please explain why there is no G power assessment to determine your overall sample size so that a accurate and predictable level of assessment can be made to give an indication of acceptability of the results??
You say in line 91: “(PHQ-9), which is an effective tool for evaluating the situation of depressive symptoms” why is this and what is it based on??
Within your covariant assessment there is no explanation on why the specific test were use, and what is the progression in the AH??
It would be of value to understand why and how you use the term mediated?? In what context are you referring to please??
Please explain, what is an inflammatory diet?? What does it consist of and what are the implications and how was it tested for??As per Pro inflammatory diet.
Line 222 you state” The hippocampus is not only an important subcortical memory structure, but also a key neuromodulator regulated the energy intake.” Please explain to what?? What is subcortical memory structure?? How does it regulate energy??
Line 260 you state” …………. mediated by different dimensions of cognitive impairment” what dimensions are you referring to and does a dimension relate to cognitive impairment??
Round 2
Reviewer 1 Report
The Authors done minimum, by changing exactly those parts which I explicitly mentioned. However, not all. E.g. Still inside Table 4 are put "95%CI" instead of values (column Mediated (Indirect Effect)). This is unclear for me.
Author Response
Reviewer Point: The Authors done minimum, by changing exactly those parts which I explicitly mentioned. However, not all. E.g. Still inside Table 4 are put "95%CI" instead of values (column Mediated (Indirect Effect)). This is unclear for me.
Response: Firstly, thank you for your suggestions which improved the quality of our manuscript. Secondly, I apologize for the confusion between P-values and 95% CIs. In the last version, the mediation analysis was performed by SPSS, which only yields 95% CI and not P-values. And the significance was usually determined based on the 95% CI without the 0 value. However, the Mplus could obtain the P-value of the Mediated (Indirect Effect), therefore, we have added the Mplus analysis in this version. We have added the P-value in the table 4 and this software in the statistical analysis section in the manuscript (line 164, in yellow). We glad that you provide the suggestions helping us to refine our manuscript. Again, we apologize that this version does not meet your requirements, and we are glad to revise our manuscript according to your suggestions.
Reviewer 2 Report
The authors have appropriately responded to and addressed my comments on the earlier manuscript
Author Response
Thank you for your patience in reviewing our manuscript and your suggestions have improved the quality of our manuscript. Wish you all the best.
Reviewer 3 Report
Dear Authors,
Thank you for taking on board the suggestions as it makes the manuscript far more specific, readable and of improved value. I have no further changes.
Author Response

(The authors gave the same response as above.)
